# The Effect of Foliar and Ground-Applied Essential Nutrients on Huanglongbing-Affected Mature Citrus Trees

**DOI:** 10.3390/plants10050925

**Published:** 2021-05-06

**Authors:** Alisheikh A. Atta, Kelly T. Morgan, Davie M. Kadyampakeni, Kamal A. Mahmoud

**Affiliations:** 1Southwest Florida Research and Education Center, University of Florida, 2685 SR 29 N, Immokalee, FL 34142, USA; conserv@ufl.edu (K.T.M.); kaem@ufl.edu (K.A.M.); 2Citrus Research and Education Center, 700 Experiment Station Rd, Lake Alfred, FL 33850, USA; dkadyampakeni@ufl.edu

**Keywords:** *Candidatus* Liberibacter asiaticus, HLB, irrigation scheduling, LAI, micronutrients, soil ammonium–nitrogen, soil nitrate-nitrogen, tree canopy volume, vegetative growth

## Abstract

The fate of foliar and ground-applied essential nutrients is the least studied topic under citrus greening or Huanglongbing (HLB)-affected citrus, which is inherently suffering from severe root decline because of HLB-associated problems. The objective of this study was to evaluate if ground-applied coupled with foliar spray of essential nutrients can reverse the decline in tree growth and understand the fate of the nutrients in the soil-root-tree interfaces. The treatments were arranged in a split-split plot design in which nitrogen (N) was ground-applied in 20 splits biweekly and Mn, Zn, and B were foliar and /or ground-applied in three splits following the spring, summer, and late summer flush seasons. Soil nutrients in three depths (0–15, 15–30, and 30–45 cm), root, and leaf nutrient concentrations of the essential nutrients, leaf area index (LAI), and tree canopy volume (TCV) data were studied twice (spring and summer) for two years. A significantly higher soil NH_4_-N and NO_3_-N concentrations were detected in the topsoil depth than the two lower soil depths (15–30 and 30–45 cm) indicating lesser nutrient leaching as trees received moderate (224 kg ha^−1^) N rate. Except for soil zinc (Zn) concentration, all the nutrient concentrations were significantly higher in the topsoil (0–15 cm), compared with two lower soil depths indicating that Zn was intricate by changes in soil environmental conditions, root acquisition, and/or leaching to lower soil depth. Leaf N concentration significantly increased over time following seasonal environmental fluctuations, tree growth, and development. Thus, leaf N concentration remained above the optimum nutrient range implying lower N requirement under irrigation scheduling with SmartIrrigation, an App used to determine the daily irrigation duration to meet tree water requirement and split fertigation techniques. Root Manganese (Mn) and Zn concentrations were significantly higher in the root tissues of the treated than the control trees and translocated to the leaves accordingly. Meanwhile, a significantly higher LAI for trees budded on Swingle (Swc) rootstock however, larger TCV for trees budded on Volkameriana (Volk) rootstocks. The trees had significantly larger TCV when the trees received a moderate N rate during early study years and under foliar 9 kg ha^−1^ coupled with the ground 9 kg ha^−1^ Mn and Zn treatments during the late study years. Therefore, split ground application of 224 kg ha^−1^ of N, foliar applied 9 kg ha^−1^ coupled with ground-applied 9 kg ha^−1^ Mn and Zn were the suggested rates to sustain the essential leaf nutrient concentration within the optimum ranges and improve the deterioration of vegetative growth associated with HLB-induced problems of citrus trees.

## 1. Introduction 

The bacteria *Candidatus* Liberibacter asiaticus are Gram-negative, phloem-limited, the primary cause of Huanglongbing disease (HLB, citrus greening), fast-proliferating, and the most destructive disease of citrus trees since its detection in Florida in 2005 [1,2,3,4]. The Asian citrus psyllid (*Diaphorina citri Kuwayama*), the disease’s vector, is a widespread pest throughout Florida [3,5,6,7]. Once a citrus tree is infected with HLB, it becomes nonproductive within a few years [8,9]. HLB alters citrus tree physiology (photosynthesis and xylem flux) and morphology (e.g., phloem translocation, bloated and corky veins, root length, and density) subsequently affecting nutrient acquisition, movement, and utilization [10,11]. Several methodologies were devised that include soil acidification, enhanced nutritional programs, split fertigation, and foliar spray of essential nutrients to reverse the decline of citrus trees as the result of HLB-induced citrus deterioration [7,12,13,14,15].

Foliar nutrition can complement ground-applied fertilizers to satisfy crop nutrient demand, particularly when citrus root systems are restricted, in soil having a severe nutrient deficiency, nutrients immobilized because of undesirable soil conditions [7,13,16,17,18]. Nevertheless, the foliar spray is not expected to entirely replace or eliminate ground-applied N, P, and K fertilization [17,19]. Foliar applied nutrients alone are inadequate to support normal tree growth [7,20]. Foliar applications of some micronutrients, Zn, Mn, and B, are an efficient method for satisfying citrus nutrient needs [18,21]. Ground-applied fertilizers could be exposed to detrimental soil processes, such as immobilization to plant unavailable forms, leaching, and runoff. Foliar spray of micronutrients Mn, Zn, Cu, B, and Mo, are significantly more efficient and economically viable than soil applications if applied in sufficient quantities [22,23]. 

Nitrogen is one of the essential nutrients for citrus production that play a vital role in canopy growth and fruit yield [14,24]. Florida sandy soils coupled with leaching rain events and extended irrigation duration exacerbate the possibility of N leaching beyond the root zone [12,25,26,27]. Acidic and alkaline soils are common problems in Florida citrus groves because of leaching in the acidic soils and precipitation in alkaline soils [18,28]. Some essential nutrients (e.g., Ca, Mg, Fe, Mn, Zn, and B) are less available for uptake by some crops under alkaline soil conditions [28,29]. Therefore, multiple annual fertilizer applications in association with scheduling fertigation, and products with incorporated elemental sulfur (S) that releases S to lower soil pH were utilized to enhance nutrient availability and minimize the nutrient leaching probability [28,30]. Manganese (Mn) deficiency is normally a worldwide problem predominantly manifested in the spring after a cold winter [2,31]. Severe Zn or Fe deficiencies bring about a delay in recognizing the symptoms of Mn deficiency and subsequent corrective measures. Soils with insufficient levels of the essential plant micronutrient Zn are common worldwide [7,32]. In a study conducted on the perennial plant *Phytolacca acinosa*, Mn accumulation in tissues (leaves, stems, roots) increased with time with an increase in the supply of the amount of Mn [33,34]. Similarly, partitioning of Mn was found between shoots and roots, displaying the ability of Mn movement from roots to shoots, remarkable tolerance to Mn but also has greater uptake and accumulation [15,34]. Mn deficiency increases the vulnerability of plants to pathogen attack [31,35]. Though citrus trees possess a thick cuticle leaf layer, they react adequately to foliar nourishment because of numerous and wider stomata on the abaxial surface coupled with a high quantity of cuticle apertures than other epidermal cells [19,36]. On the other hand, the peak photosynthetic activity is manifested in mature leaves, the source of photosynthetic produce for the remaining parts of the plant parts. At that phase, foliar-applied nutrition is absorbed and translocated from leaves to flowers, fruits, and roots along with assimilates [13,19]. 

The physiological impact of Zn on the leaves and fruit have not been identified [13,18,30,32]. Phloem translocation of Zn is one of the important features that contribute to HLB-induced Zn deficiency in grapefruit and the concentration was too low in young and mature leaves, such that no clear Zn distribution was detected in the phloem tissues [4,32]. Research showed that there is indirect evidence for high Zn translocation in the phloem of healthy citrus, which previously had been associated with intermediary phloem mobility [18,23,32]. Physiological, morphological, and biochemical abnormalities are usually initiated by Zn deficit resulting in a decline in flowering, fruit set, and yield losses [4,18,37]. Boron (B) is a vital micronutrient but the least understood in plant physiology [11,38]. A decrease in leaf B has been associated with HLB-affected citrus trees [11,38,39]. Inherently, Florida sandy soils are low in B, and deficiency of B is exacerbated by extreme drought, high pH irrigation water, or excessive lime dressing [29,37]. Soil pH is an imperative factor that influences the availability of B in soil and its availability progressively decreases with increasing soil, pH particularly beyond soil pH 6.5 [39,40]. Boron takes part in carbohydrate translocation through the phloem, cell wall structure, and influences pathogen susceptibility [11]. Soil pH is an imperative factor that influences the availability of B in soil and its availability progressively decreases with increasing soil pH, particularly beyond soil pH 6.5 [37,38,39]. Boron takes part in carbohydrate translocation through the phloem, cell wall structure, and influences pathogen susceptibility [11,38,39]. The impact of rootstocks on scion performance, micronutrient rate, type, method of applications on HLB-affected citrus trees have not been well addressed. The fate of essential nutrients in the soil-root-tree interfaces of HLB-affected citrus trees is not well studied [22,41,42]. Therefore, the following hypotheses were outlined: (1) split foliar and /or ground-applied essential nutrients improve soil availability, accumulation, and uptake of nutrients by HLB-affected citrus trees; (2) split application of essential nutrients reduces nutrient leaching and the detrimental effect of soil immobilization as result of soil reactions; (3) split application of essential nutrients increases the nutrient concentration in the soil, root, and leaf tissues; and (4) adding essential nutrients boost LAI and TCV of HLB-affected citrus trees. Therefore, the objective of this study was: (1) determine if split applications of essential nutrients increase soil N, Mn, Zn concentration over time; (2) compare foliar and ground-applied essential nutrients improve the root and leaf N, Mn, and Zn concentrations of HLB-affected citrus trees; (3) evaluate if ground applications in addition to the foliar spray of essential micronutrients (Mn and Zn) and split ground application of N reverse the decline in vegetative growth, such as LAI and TCV. 

## 2. Materials and Methods

### 2.1. Study Site, Experimental Design, and Treatments

Mature sweet orange trees (*Citrus × sinensis* L. Osbeck cv. “Valencia late” budded onto Volkameriana (Volk) lemon (*C*. *volkameriana* Pasquale) or Swingle (Swc) citrumelo (*Citrus × paradisi Macf. × Citrus trifoliata* L. Raf.) rootstocks were planted in 2006. The test was conducted at the University of Florida, Southwest Florida Research and Education Center near Immokalee, FL from 2017 to 2019. The soil at the test site was classified as Immokalee fine sand (sandy, siliceous, hyperthermic Arenic Alaquods). A randomly selected mature five leaves per tree were collected and processed for quantitative polymerase chain reaction (qPCR) analysis in HLB diagnostic laboratory located at Southwest Florida Research and Education Center (Immokalee, FL) in February 2017. Leaf samples with cycle threshold (Ct) values of 30 or less were deemed HLB-positive, with Ct values of more than 32 or more were HLB-negative. Hence, foliar laboratory results of the HLB-affected trees under the study showed a cycle threshold value of 24.7 ± 0.19, indicating an active infection or more copies of the CLas bacteria were amplified during each cycle. 

The test comprised of three main effects as the main block of two rootstocks, intermediary subplots of three N rates, and sub subplots of three micronutrient rates in which each sub subplot consisted of 0.036 ha. The treatments were assigned in random in each experimental units where the three N rates (168, 224, and 280 kg ha^−1^) were split within the large block of the rootstocks, the sub subplots which received either 0× (control), 1× (foliar only), 2× (1× foliar and 1× ground-applied), or 3× (1× foliar and 2× ground-applied) bentonite clay encapsulated micronutrient product containing elemental sulfur and Mn and Zn oxide, as previously suggested by the University of Florida, Institute of Food and Agricultural Sciences for citrus nutrition [43]. Each of 1× was equivalent to 9 kg ha^−1^ of S coated metallic Mn and Zn oxides and 2.3 kg ha^−1^ of B nutrients (Table 1). Meanwhile, all trees received K fertilizer (as K_2_O) at 168 kg ha^−1^. 

The N and K were fertigated as split applications on a biweekly basis between February and November in each study year. Trees were fertigated with a micro-sprinkler Max−14 (Maxijet Inc., Dundee, FL, USA), Fill-in blue emitters were placed 15 cm from the tree trunk. The irrigation was assisted with a 172.4 kpa irrigation pump at a flow rate of 45 L h^−1^ [30]. Citrus SmartIrrigation app, which is available on iOS and Android interfaces, was used to determine daily irrigation duration to meet tree water requirement based on reference evapotranspiration (ET_o_) calculated with Penman-Monteith and crop coefficient related to the meteorological data of the study site [12,30].

### 2.2. Soil Sampling, Extraction, and Nutrient Analysis

Soil samples were collected within a 0.5–1.0 m radius under the tree canopy. The samples were collected at three different locations along with three soil profiles: 0–15, 15–30, and 30–45 cm per tree in spring (Feb. or Mar.) and summer (Aug. or Sept.) of 2017 and 2018 growing seasons. The collected samples were stored in a freezer until soil extraction [10,30]. Soil NH_4_-N and NO_3_^−1^-N concentrations were extracted using the 2M KCl by pipetting a 40 mL of 2M KCl solution into an extraction tube containing approximately 5.0 g (±0.05) of wet soil, capped, and shaken for 30 min on a reciprocating shaker [44]. Subsequently, the supernatant was left to settle for 15 min, filtered using Fisher Scientific filter paper placed onto labeled vials, capped, and kept in a freezer until analysis [45]. The soil NH_4_-N and NO_3_-N concentrations were processed using a Microplate Reader spectrophotometric (Epoch BioTec, Winooski, VT, USA) at 660 and 520 nm, respectively, and calculate on a dry weight basis [12,46].

A 25 mL of Mehlich III extractant solution (0.2 M CH_3_COOH + 0.015 M NH_4_F + 0.013 M HNO_3_ + 0.001 M EDTA + 0.25 M H_4_NO_3_) was pipetted into an extraction tube containing dry soil samples of 2.5 g (±0.05) [46,47] to extract Mehlich III-extractable concentrations of P and heavy metals. The sample was capped and shaken for 5 min at a high rate (200 reciprocating per minute). Subsequently, the supernatant was allowed to rest for 30 min, after which the solution was filtered onto labeled vials, and stored in a freezer pending analysis [12,14,48]. Mehlich III-extractable concentrations of P and heavy metals were processed using inductively coupled plasma optical emission spectroscopy (ICP-OES, Spectro Ciros CCD, Fitzburg, MA, USA). All results were expressed on an oven-dry soil weight basis [12,46].

### 2.3. Leaf and Root Tissue Nutrient Concentration

Leaf and root samples were collected in the spring (February/March) and summer (August/September) seasons. Twenty leaves (4–6 month-old) were collected from non-fruiting branches [43]. The leaf samples were washed in a weak detergent solution (0.1–0.3%, Micro-90 ®Burlington, NJ, USA), rubbed between the thumb and forefinger, subsequently rinsed with reverse osmosis water followed by deionized water to remove nutrients adhering to the leaf surface [34]. Approximately 50 g of fine roots of < 2.0 mm were also collected from the topsoil depth (0–15) cm using a hand rake across the half canopy on the swale side of the bedded rows. The root samples were gently shaken to remove the soil, kept in a labeled kaki paper bag, and brought to the processing lab for cleaning the roots. The samples were then oven-dried at 65°C, before grinding to pass through a 40 mesh screen in a Wiley mill (Thomas-Wiley Laboratory Mills, Philadelphia, PA, USA) [49,50,51]. Oven-dried sample (0.5 g) was weighed, processed [43], and stored in a refrigerator at < 4 °C pending analysis with Inductively Coupled Plasma Optical Emission Spectroscopy (Spectro Ciros CCD, Fitzburg, MA) [7,52,53]. The NA2500 carbon (C)/N Analyzer (Thermoquest CE Instruments; ThermoQuest Corporation, Thermo Fisher Scientific Inc., Waltham, MA, USA) was used to determine tissue N (%). Subsequently, the leaf nutrient concentration was compared with the critical nutrition concentration that was previously established from years of experimentation [43].

### 2.4. Vegetative Growth

#### 2.4.1. Leaf Area Index

Canopy density or leaf area index (LAI) is an indicator of above-ground vegetation growth. The measurement of LAI was determined by the amount of radiation transmission through the tree canopy using the radiative transfer principle [54] with a SunScan canopy sensor system (Dynamax Inc., Houston, TX, USA) at mid-day (1130 to 1330 HR), in which the solar zenith angle was within < ±10° (±0.1) [12]. The average of four readings per tree was made across the four directions: east, west, north, and south from three sub subplots and replicated four times (*n* = 48).

#### 2.4.2. Tree Canopy Volume

The average canopy diameter along the east-west and north-south and canopy height was measured to calculate the canopy volume using Equation (1). The TCV was measured at the beginning (Feb. or Mar.) of the experiment and every six months thereafter (Feb. or Mar. and Aug. or Sept.) for three years.(1)$$TCV=\frac{4}{3}\times \left(\pi \right)\left(r^{2}\right)\times \left(\frac{h}{2}\right)$$
where *TCV* (m^3^) = Tree canopy volume; *r* = mean canopy radius (m); *h* = canopy height (m).

### 2.5. Statistics and Data Analysis

Three-way interaction effect among the main factors: rootstock, N rates, and micronutrient rate treatment effects was tested on soil, root, and leaf nutrient concentration, LAI, and TCV. Field data on rootstocks, N rates, micronutrients were deemed as fixed effects, while replication in sub and sub-sub blocks were assigned as the random effect to the PROC General Linear Model (GLM) procedures SAS 14.1 (SAS Institute, Cary, NC, USA). This allowed us to see all the main effects and interaction effects between rootstocks, N rates, and micronutrient rates. The data were tested for linearity, normality, homogeneity of variance, and independent errors before the derivable statistics analyses were performed. Mean separation among the means with a statistical difference at the *p*-value of ≤ 0.05 was performed using the Tukey–Kramer honestly significant difference grouping range test. All graphs were formed using the Sigma Plot 14 (SigmaPlot 14, Systat Software, San Jose, CA, USA).

## 3. Results and Discussion

### 3.1. Soil Nutrient Concentration

#### 3.1.1. Soil Nitrate Nitrogen

During the entire study seasons, the amount of soil NO_3_-N was higher on the topmost depth (0–15 cm) as compared with the two lower soil depths (15–30 and 30–45 cm) regardless of the N rates (Table 2 and Figure 1A–C). The amount of soil NO_3_-N significantly increased at the lowest soil depth (30–45 cm) with the moderate N rate and was lowest with the lowest N rate (168 kg ha^−1^) (Figure 1B). Meanwhile, higher soil NO_3_-N concentration was detected when trees received 224 kg ha^−1^ than 280 kg ha^−1^ N during summer 2018 indicating that more soil NO_3_-N leaching to the lower soil depth or more N was absorbed by the trees on the highest N rate, as compared with the lowest N rate. Higher soil NO_3_-N concentration in the lower soil depth as a result of soil NO_3_-N leaching following leaching rain events of Hurricane Irma (300 mm day^−1^) in summer 2017. The relative accumulation and leaching in soil NO_3_-N concentration during 2017 was not conspicuous along with the soil profile and within the N rates. During spring 2018, the soil NO_3_-N showed an increase of 7%, 30%, and 23% of soil NO_3_-N concentration in the topmost soil depth as compared with its immediate soil depth pertained to 168 kg, 224 kg, and 280 kg ha^−1^, respectively. In summer 2018, 36% more soil NO_3_-N was accumulated in the topmost soil depth for trees that received 200 kg ha^−1^ than 168 kg ha^−1^.

During summer 2018, the soil NO_3_-N showed an increase of 9%, 42%, and 65% of soil NO_3_-N concentration in the topmost soil depth as compared with its immediate soil depth ascribed to 168 kg, 224 kg, and 280 kg ha^−1^, respectively. The overall range of the soil NO_3_-N concentration varies between 0.3–4.0 mg kg^−1^ throughout the study season. These findings are similar to those results documented under similar fertigation programs on HLB-affected Hamlin citrus trees with optimum irrigation scheduling [12,17,55]. In a previous study in conventional N application methods, under a mature sweet orange tree, on well-drained soil with optimum irrigation, and regular split fertigation ranging from 112 to 280 kg ha^−1^ N the soil NO_3_-N concentration was 8–15 mg kg^−1^, which was a value of 200 to 400% more than the current study [12,17,22,56]. Therefore, irrigation scheduling and multiple split application of N utilized in the current study was per the best management practices (BMP) directed by the Florida Department of Agriculture and Consumer Services for sustainable citrus production in the era of HLB which is severely affecting the citrus industry [29,57].

#### 3.1.2. Soil Ammonium Nitrogen

Like the soil NO_3_-N, the soil of NH_4_-N concentration was significantly higher in the topsoil depth as compared with the two lower depths regardless of the N rates during the 2018 seasons (Table 2 and Figure 1D–F). The accumulation soil NH_4_-N concentration in the topmost soil depth as compared with its immediate lower soil depth was a value of 9%, 30%, and 10% greater in the spring and 10%, 40%, and 7% greater in the summer of 2018 under 168, 224, and 280 kg ha^−1^ N rates, respectively. None of the N rates showed a significant variation to the movement of soil NH_4_-N along with the soil profile irrespective of the N rates. This indicated that the dynamic of soil NH_4_-N to the lower soil depth was limited as compared with the soil NO_3_-N concentration. Similar results were reported in which the magnitude of soil NH_4_-N ranging from 1.0 to 6.0 mg kg^−1^ in the topsoil depth under restricted and drip irrigations with poorly drained soils [22,48]. Meanwhile, the restricted soil NH_4_-N mobility along the soil profile and the conversion of 33 to 41% of the applied NH_4_-N to NO_3_-N in 7 days could be the best strategy for year-round availability of N to the citrus trees suffering from severe deterioration of fine root density from HLB-associated problems [12,22,58,59].

#### 3.1.3. Soil Manganese Concentration

Early in the study season, the soil Mn concentration reacted according to the amount of Mn applied foliar and to the ground among the three soil depths (Table 2 and Figure 2A–C). The soil Mn concentration was significantly higher only when trees received the foliar (1×) and ground (1×) or foliar and the ground (2×) doses, while only the foliar treated trees showed no significant difference as compared with the control trees. Following the spring 2017 leaf flushes, the acquisition of Mn from the soil pool and the translocation of Mn from accumulated tissues to the newly growing tissues created a potential gradient in the leaves to absorb more nutrients, which were discussed below. The soil Mn concentration in the soil significantly increased over time when trees received the foliar (1×) couple with the ground (1×) as well as foliar (1×) couple with the ground (2×) Mn and Zn treatments. However, since foliar Mn spray promotes above-ground vegetative biomass, the foliar spray could significantly exploit more of the soil Mn pool as compared with the control trees on the topmost soil depth (Figure 2A). The soil Mn concentration was significantly lower in the two lower soil depths under the control trees, proving that there was deteriorating soil Mn concentration because of no external source to replenish Mn acquisition by crop removal.

#### 3.1.4. Soil Zinc Concentration

Soil Zn was less likely to accumulate in the topsoil depth. It was uncertain if it moved to the lower soil depth or absorbed by the root and moved to leaf tissues. During the early seasons, no significant variations of soil Zn concentrations were detected in the topmost soil depth (Figure 2D–F). However, later in the seasons, more soil Zn concentrations were observed gradually to accumulate in the lower soil depth (Figure 2E,F). Ground-applied Zn is less accessible to plants root because of its low movement, high soil fixation, and HLB-induced root injury [16,18]. Since no proof was reported to recover Zn deficiency by adding to the soil or immobilized zinc within the plant tissue, foliar spray of Zn is a suggested method to increase leaf Zn tissue concentration in plants [18,60].

#### 3.1.5. Other Soil Nutrients

A significant effect on soil Ca, Mg, and NH_4_-N was detected as the result of adding Mn and Zn in the soil consistently during the spring and summer seasons (Table 3). Soil Ca concentration showed a magnitude of 1.9×, 3.3×, and 1.9× in the spring and 1.5×, 1.6×, and 3.5× lesser in summer under the control trees in response to the foliar (1×), foliar (1×) couple with the ground (1×) and foliar (1×) couple with the ground (2×), respectively. Similarly, Soil Mg concentration showed a magnitude of 1.3×, 2.5×, and 4.0× in the spring and 1.0×, 0.8×, and 3.3× lesser in summer under the control trees attributed to the foliar (1×), foliar (1×) couple with the ground (1×) and foliar (1×) couple with the ground (2×), respectively. Similar metallic elements have an antagonistic effect, can replace, and compete for the same plant acquisition, movement, and storage site within an organism [61]. In previous studies indicated that any divalent element impedes the acquisition of other elements such as Mn, Zn, Ca in the soil, which might induce reduced uptake and toxicity of other cations [4,12,61]. On the other hand, soil NH_4_-N increased with increasing the application of Mn and Zn. Soil NH_4_-N had 1.1×, 1.6×, and 2.3× greater than the control trees that received foliar (1×), foliar (1×) couple with the ground (1×) and foliar (1×) couple with the ground (2×), respectively. Soil Cu and Fe concentrations were not significantly affected on the topmost depth soil in reaction to the ground-applied essential nutrient treatments in neither of the study seasons.

### 3.2. Root Nutrient Concentration

#### 3.2.1. Root Mn Concentration

The root Mn concentration remained significantly higher for trees that received foliar (1×) coupled with the ground (1×) and foliar (1×) coupled with the ground (2×) treatments as compared with control trees in the spring season (Figure 3A,B). Soil Mn concentration showed a magnitude of 1.2×, 2.6×, and 2.4× greater for trees budded on Volk and 1.7×, 3.8×, and 4.4x greater for trees budded on Swc the spring and 2.1 ×, 3.1×, and 4.3× greater for trees budded on Volk and 1.7×, 3.8×, and 4.4x for trees budded on Swc rootstocks in the summer than the control trees ascribed to the foliar (1×), foliar (1×) couple with the ground (1×), and foliar (1×) couple with the ground (2×), respectively. The root Mn concentration showed no significant difference between foliar (1×) treated and the control trees during the spring and summer seasons. Similarly, foliar (1×) coupled with the ground (1×) and foliar coupled with the ground (2×) Mn and Zn treatment showed no significant variation in root Mn concentration. This indicated that only foliar spray is not enough to show variation in root Mn concentration compared to the untreated trees and foliar (1×) and ground (2×) application was less reactive to the excess application compared with foliar (1×) and ground (1×) Mn applications. However, these premises will not provide us the notion of tree optimum Mn concentration unless the accumulated Mn nutrition in the root is showed up in the leaf nutrient concentration, which later will be discussed in the leaf nutrient concentrations. Besides, it was documented that Zn, Mn, and Fe are among the most micronutrients affected by HLB-associate problems, particularly in the root tissues of HLB-affected citrus trees [15]. Generally, foliar and /or ground-applied Mn nutrients increase root Mn concentration approximately 35–65% greater in the foliar (1×) treated trees and 60–85% more in the foliar (1×) coupled with the ground (1×) treated trees as compared to the control trees. These fine roots of Valencia Mn concentration grown on sandy soil were similar to those reported by [62], 250 mg kg^−1^.

#### 3.2.2. Root Zinc Concentration

Foliar (1×) coupled with the ground (1×) treated trees showed a significantly higher root Zn concentration compared with the control trees (Figure 4A,B). No significant variation of root Zn concentration was detected between the foliar (1×) and untreated trees implying a required soil Zn application. Soil Zn concentration showed a magnitude of 1.8×, 2.1×, and 1.5× greater for trees budded on Volk and 1.6×, 1.9×, and 3.1x for trees budded on Swc in the spring and 1.6×, 1.9, and 3.1× for trees budded on Volk and 1.3×, 2.1×, and 1.8x for trees budded on Swc rootstocks in summer under the control trees ascribed to the foliar (1×), foliar (1×) couple with the ground (1×), and foliar (1×) couple with the ground (2×), respectively. There was no significant noticeable difference of root Zn concentration between the foliar (1×) coupled with the ground (1×) and foliar (1×) coupled with the ground (2×) treated trees in any of the seasons implying with increasing Zn soil application would result in an antagonizing effect with other divalent metallic elements and soil fixation problems discussed earlier. Previous results reported that competition of Zn with other divalent mineral nutrients like Fe, Mn, and Ca result in Zn deficiency [21,53,61]. No significant difference in root Zn concentration was also detected in trees underground and foliar treated trees regardless of rootstock type, nutrient doses, and method of Zn application. Foliar Zn spray was reported to be less effective than ground-applied Zn in increasing vegetative growth, which eventually cannot be manifested to translocate from the top to down to correct Zn deficiency in the roots [18]. Meanwhile, the current study indicated that extra Zn application beyond the ground (1×) dose showed unnecessary during the entire season. A previous study indicated that reduced phloem dynamics of Zn is the primary factor that is prevalent on HLB-driven Zn insufficiency in grapefruit [32]. The pre-HLB era indicated lower fine root micronutrient concentration of Valencia citrus trees grown on sand soil, (mg kg^−1^): Mn = 250; and Zn = 409 supporting the aforementioned statements [62].

### 3.3. Vegetative Growth and Leaf Nutrient Concentration

#### 3.3.1. Leaf Nutrient Concentration

During the first year, the leaf N concentration remained below the optimum range concentration regardless of the rootstock type and N rates (Figure 5A,B). In spring 2017, the Leaf N concentration was significantly higher for the lowest (168 kg ha^−1^) and highest (280 kg ha^−1^) N rates (Figure 5A). The high leaf N concentration in the lowest N rate probably could be because lowest TCV in which the leaf N concentration appeared to be higher. Meanwhile, the high leaf N concentration in the highest N rate could also be because of the highest N amount received by the trees. Later in the study years, the 224 kg ha^−1^ N showed significantly higher N concentration as compared with the highest and lowest N rates. However, no significant leaf N concentration was noted for trees budded on Volk rootstock irrespective of the N rate suggesting a lower N requirement. As will be discussed later in this paper regarding TCV, with increasing N beyond the 224 kg ha^−1^, the trees did not respond to the addition of N with differences in TCV. Adding less than 224 kg ha^−1^ N limited the potential tree growth further. This indicates that leaf N concertation was elevated by the split N application that made the N available and taken up by the trees throughout the season. To connect the dots, the 200% to 400% less soil NO_3_.-N in the current split application than in the conventional N application could be the plausible reason for the elevated leaf N concentrations. Similar or less leaf N concentration was reported on fleshy fruits in which higher N concentration was detected in the early stage of fruit development and decline progressively through time as the flush season and fruit growth and maturity advances because of the dilution effect inflicted by N translocation from old to the young leaves, scion to rootstock association, weather conditions, and agronomic practices [36,48]. Dilution is the decrease in nutrient concentration created as the result of translocation of nutrients from mature leaves and to the newly growing shoots and roots [7,48,63]. The allocation of nutrients to the increased vegetative growth resulted in a decrease in the nutrient concentrations in trees with higher vegetative density (LAI) and TCV discussed below. Therefore, increased biomass and N distribution relationships established on tree size measurements revealed the efficiency of the nutrition acquisition when the tree has higher above-ground biomass [24,27,52].

*Leaf Mn concentration*. As with soil Mn, leaf Mn concentration was reactive to the amount of Mn applied to either the ground and/or foliar. Foliar (1×) coupled with the ground (1×) or foliar (1×) coupled with the ground (2×) treatments constantly showed above the optimum nutrient ranges (Figure 6A). During peak growing season in summer 2017, leaf Mn concentration was 3.8×, 4.4×, and 6.8× greater than the control trees ascribed to foliar (1×), foliar (1×) couple with the ground (1×) and foliar (1×) couple with the ground (2×), respectively. The leaf Mn concentration of the control trees was consistently below, foliar (1×) within, and foliar (1×) couple with the ground (1×) and foliar (1×) couple with the ground (2×) were above the optimum nutrient concentration ranges throughout the study seasons. Thus, in summer 2018, leaf Mn concentration showed 2.7×, 7.8×, and 5.6× greater than the control trees attributed to the foliar (1×), foliar (1×) couple with the ground (1×) and foliar (1×) couple with the ground (2×), respectively. Even though, leaf Mn concentration under foliar (1×) treatments was consistently within the optimum ranges, during heavy Mn crop removal season or severe vegetative fall growth following a strong windy season could result in a decline in the leaf Mn concentration below the optimum leaf nutrient concentration ranges. Research results indicated that up to 80% lower Mn in the root tissues and 58% in the leaf tissue of HLB-affected than HLB-free citrus trees [15,20,32]. Meanwhile, the increase in above-ground vegetative growth exploited the soil Mn pool in the soil and inflicted a rapid decrease in leaf Mn concentration because of the dilution effect.

*Leaf Zinc concentration.* Leaf Zn concentration was significantly higher in the Mn and Zn treated trees than the control trees except for the data collected after Hurricane Irma. In summer 2017, leaf Zn concentration of the treated trees were 3.8×, 4.0×, and 4.1× greater than the control trees associate with foliar (1×), foliar (1×) coupled with the ground (1×), and foliar (1×) coupled with the ground (2×), respectively (Figure 6B). Similar results were reported for leaf Zn concentration ranging 25–31 (mg kg^−1^) on Hamlin, Parson Brown, Valencia, and Sunburst Citrus varieties under the control trees [63]. The sharp decrease of leaf Zn concentration in the treated trees noticed in spring 2018 could be because of heavy defoliation instigated by Hurricane Irma in September 2017. Heavy crop removal from the soil during summer and followed by fruit harvest, Zn soil fixation problems, Zn leaching beyond the root zone, and defoliation of leaves, branches, and twigs are the plausible justification for the sharp drop in the leaf Zn concentration. This is indirect evidence of the high mobility of leaf Zn accumulated in the source to the newly vegetative growth points [15,34]. However, the treated trees showed an increment in leaf Zn concentration following the spring and summer Zn application; yet, no significant difference among the treated tree was another indication of less mobility of Zn from the soil to the leaf in addition to the high soil Zn fixation and root injury in HLB-affected citrus trees [16,18].

#### 3.3.2. Leaf Area Index

In summer 2017, a significantly higher LAI for tree budded on Swc than Volk rootstocks was detected attributed to nutrient effect (Table 4). The LAI results documented after Hurricane Irma in summer 2017 indicated that an overall loss of 2%, 17%, 3%, and 19% for trees budded on Volk and 30%, 29%, 24%, and 33% for trees budded on Swc rootstocks in reaction to the control, foliar (1×), foliar (1×) coupled with the ground (1×) or foliar (1×) coupled with the ground (2×) treated trees, respectively. The resistance to defoliation to Hurricane Irma coupled with the big canopy frame of the trees budded on Volk rootstocks resulted in the loss of the entire tree blocks. However, in summer 2018 the treated trees showed an increment in LAI of 5.5%, 5.5%, and −16% associated with foliar (1×), foliar (1×) coupled with the ground (1×) or foliar (1×) coupled with the ground (2×) for trees budded on Swc rootstocks, respectively. Similarly, in summer 2019, 11.4%, 14.3%, and −25% increase in LAI was noted accredited to the trees that received foliar (1×), foliar (1×) coupled with the ground (1×) or foliar (1×) coupled with the ground (2×), respectively. 

A value of 6.4–8.1 of LAI has been reported in response to N-application on 22-year-old Shamouti orange trees, in which the trees produce a higher leaf density to excess N (660 g N as NH_4_-NO_3_) application [64]. Similar results under conventional fertilization, 3.7–4.8 LAI in response to different quantities of SmartIrrigation treatments was reported on 5-year-old sweet orange trees located at three commercial groves at Arcadia, Avon Park, and Immokalee, FL. Each [65]. The decrease in LAI could be because the highest Mn acquisition resulted in reducing the growth tip associated with overdose toxicity. A decrease in soil pH as the result of the S emanated from the sulfur-coated Mn and Zn products created acidic soil environment could be another reason to reduce the fine root density [30] and LAI of the trees that received foliar (1×) coupled with the ground (2×) treatments. A quadruple Mn application (4×) rate resulted in Mn toxicity that led to reduced Mn acquisition on HLB-affected Valencia citrus trees [15] a decline of biomass and photosynthesis, and biochemical disarrays associated with oxidative stress under acidic soil environments [33,66]

#### 3.3.3. Tree Canopy Volume

The TCV was significantly affected by the interaction of rootstock × N rate during the early growing seasons (Table 5). Results indicated a significantly larger TCV for trees budded on Volk than on Swc rootstocks that received the two lower N (168 and 224 kg ha^−1^) rates. These results indicated that trees budded on both rootstocks responded to increasing N rates. However, increasing N rates beyond the highest N rate (224 kg ha^−1^) eliminated variation in TCV among the two rootstocks. A linear relationship between N application and tree biomass was observed implying a persistent tree above-ground biomass, which led to the increasing in tree size rather than tree age [24]. Meanwhile, Hurricane Irma imposed a severe TCV orientation and more TCV loss was observed for trees budded on Swingle than on Volk rootstocks. Hurricane Irma, one of the strongest and lengthiest hurricanes in the Atlantic Ocean; with wind speeds of 88 KPH and rainfall of 300 mm were reported at Immokalee, FL on 10 September 2017. The TCV losses on Volk rootstocks were 12.0%, 11.6%, and 15%, while those on Swingle rootstock were 24.9%, 25.3%, and 24.6% attributed to 168, 224, and 280 N Kg ha^−1^, respectively. Consequently, the higher wind resistance of the citrus trees budded on Volk rootstocks was eliminated from the grove because of the severe damage inflicted by the Hurricane. It was only during the third year, the trees responded to the micronutrient for TCV. 

A significantly higher TCV was recorded for trees that received foliar (1×) coupled with the ground (1×) as compared with the rest of the treatments. Thus, as compared with the control trees, TCV showed an increase of 9.6%, 17.6%, 8.6% in spring 2019 and 8.0%, 20.8%, 4.3% in summer 2019 in response to foliar (1×), foliar (1×) coupled with the ground (1×) or foliar (1×) coupled with the ground (2×) treatments, respectively. TCV with high light intensity for peak net photosynthesis can be utilized to evaluate crop growth, which eventually produces fruit in proportion to the TCV [67]. An increase in TCV was detected when trees received different N rates during the early growth stage, but fewer studies were documented on the effect of micronutrients on HLB-affected citrus trees [7,68]. The current study indicated that no significant increase in TCV was detected as trees receive beyond the 224 N kg ha^−1^ coupled with foliar (1×) coupled with ground (1×) treatments. Similar results were observed, in which TCV increment was reported when trees received the suggested N rate of 224 kg N ha^−1^ and foliar Zn nutrition as compared with other N rates and application methods [31].

## 4. Conclusions

Irrigation scheduling using SmartIrrigation coupled with the split application of essential nutrients showed a significant amount of the ground-applied nutrients to retain in the topmost soil depth. The acquisition of Mn and Zn by the roots and their concentration in the root and leaf tissue was in line with the dose of the application. Increasing the rate of N to 224 kg ha^−1^ increases the leaf N concentration, LAI, and the TCV. Yet, increasing the N beyond the moderate N rate did not significantly impact TCV inferring a lower N requirement. Adding beyond 9 kg ha^−1^ of each foliar and ground applications of Mn and Zn treatments did not significantly increase soil, root, and leaf Mn and Zn concentrations, LAI, and TCV. Hence, 224 kg ha^−1^ of N and foliar 9 kg ha^−1^ coupled with ground-applied 9 kg ha^−1^ Mn and Zn were the suggested doses of essential nutrients to fulfill nutrient requirement need and reverse the decline in HLB-induced problems, while conforming to the BMPs modalities.

## Figures and Tables

**Figure 1 plants-10-00925-f001:**
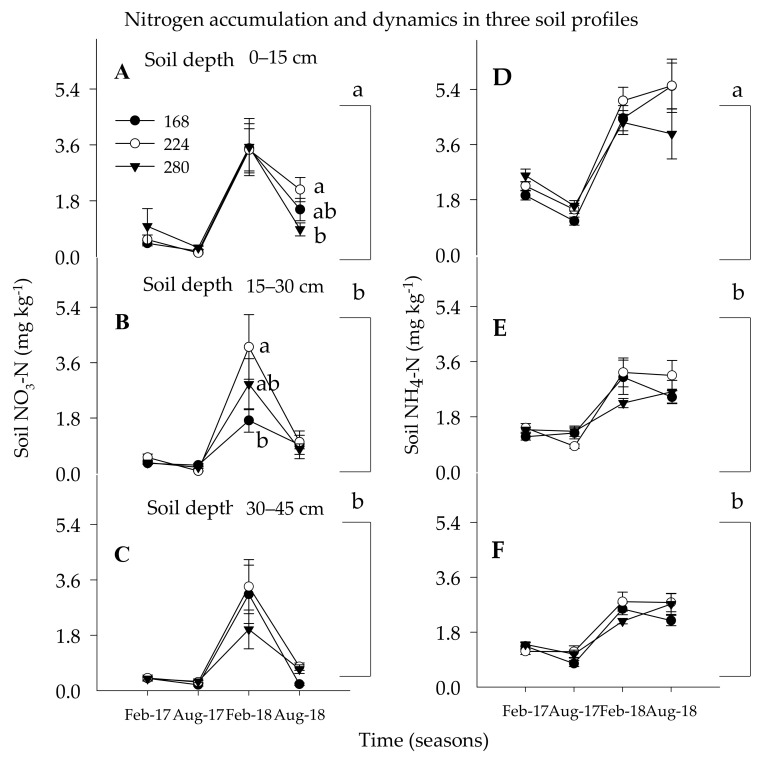
Soil available N (soil NO_3_-N: figures (**A**–**C**) and soil NH_4_-N: figures (**D**–**F**)) accumulation and dynamics under Huanglongbing-affected citrus trees as impacted by three nitrogen rate (168, 224, and 280 kg ha^−1^) in spring (sp, Feb. /Mar) and summer (su, Aug. /Sept.) 2017 and 2018 seasons in three soil depths (0–15, 15–30, and 30–45 cm). Means followed with the same lower case letters on a vertical alignment is not significantly different (*p* ≤ 0.05) using Tukey’s range test (*n* = 36).

**Figure 2 plants-10-00925-f002:**
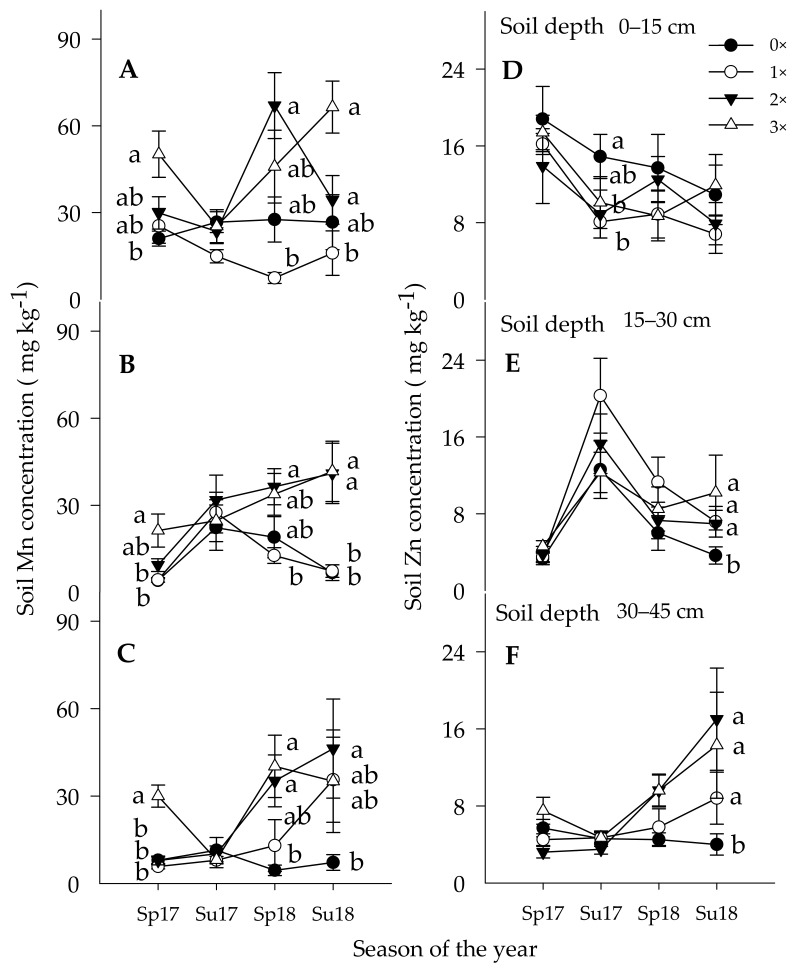
Soil Mn (figures (**A**–**C**)) and Zn (figures (**D**–**F**)) movement and accumulation in three soil depths: 0–15, 15–30, and 30–45 cm under Huanglongbing-affected Hamlin citrus trees in spring (sp, Feb. /Mar) and summer (su, Aug. /Sept.) seasons. Micronutrient treatments: control (0×, closed circles), foliar (1×) (1×, open circles), foliar (1×) coupled with the ground (1×) (2×, closed triangles), and foliar (1×) and the ground (2×) (3×, open triangles), (1× = 9 kg ha^−1^ of metallic Mn and Zn each and 2.3 kg ha^−1^ of B). Error bars denote mean ± standard error of means (*n* = 36). Means along the same vertical alignment with the same lower case letters are not significantly different (*p* ≤ 0.05) using the Tukey's range test.

**Figure 3 plants-10-00925-f003:**
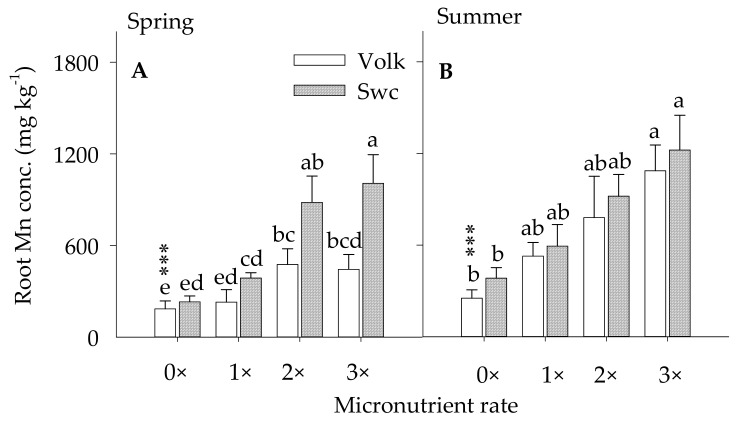
Root Mn concentration as affected by foliar and /or ground-applied micronutrients during spring (figure (**A**)) and summer (figure (**B**)) of 2017–2018. growing seasons. Micronutrient treatments (X-axis): control [0×], foliar (1×) (1×), foliar (1×) and ground (1×) (2×), and foliar (1×) and ground (2×) (3×), (1× = 9 kg ha^−1^ of metallic Mn and Zn each and 2.3 kg ha^−1^ of B). Error bars indicate the mean values (*n* = 12 trees) ± standard error of the mean for trees budded on Volkameriana (Volk) or Swing (Swc) rootstocks. Asterisks (***) indicate significance at *p* ≤ 0.0001. Means along the same vertical alignment with the same lower case letters are not significantly different (*p* ≤ 0.05) using the Tukey’s range test.

**Figure 4 plants-10-00925-f004:**
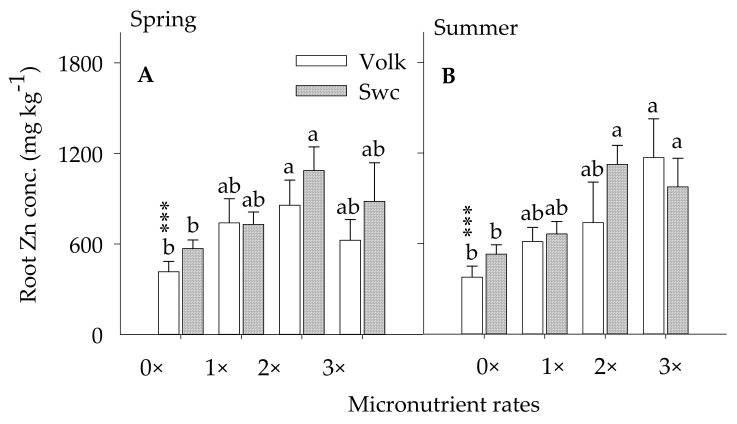
Root Zn concentration as affected by foliar and /or ground-applied micronutrients during spring (figure (**A**)) and summer (figure (**B**)) of 2017–2018 growing seasons. Micronutrient treatments (X-axis): control (0×), foliar (1×) (1×), foliar (1×) and ground (1×) (2×), and foliar (1×) and ground (2×) (3×), (1× = 9 kg ha^−1^ of metallic Mn and Zn each and 2.3 kg ha^−1^ of B). Error bars indicate the mean values (*n* = 12 trees) ± standard error of the mean for trees budded on Volkameriana (Volk) or Swing (Swc) rootstocks. Asterisks (***) indicate significance at *p* ≤ 0.0001. Means along the same vertical alignment with the same lower case letters are not significantly different (*p* ≤ 0.05) using the Tukey’s range test.

**Figure 5 plants-10-00925-f005:**
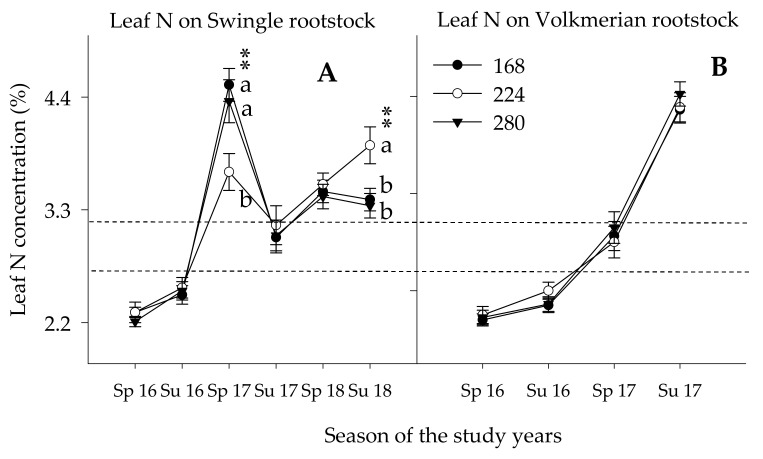
Leaf N concentration in spring (sp, Feb. /Mar) and summer (su, Aug. /Sept.) seasons of 2016–2018 growing seasons of Valencia citrus trees budded on Swingle (**A**) and Volkameriana (**B**) rootstocks at three N rates (168, 224, and 280 kg ha^−1^). No data for trees on Volk for 2018 as Hurricane Irma removed the trees in summer 2017. Asterisks (**) indicate significance at *p* ≤ 0.01. The space between the two broken lines indicates the optimum range of leaf nutrient concentrations. Means along the same vertical alignment with the same lower case letters are not significantly different (*p* ≤ 0.05) using the Tukey’s range test.

**Figure 6 plants-10-00925-f006:**
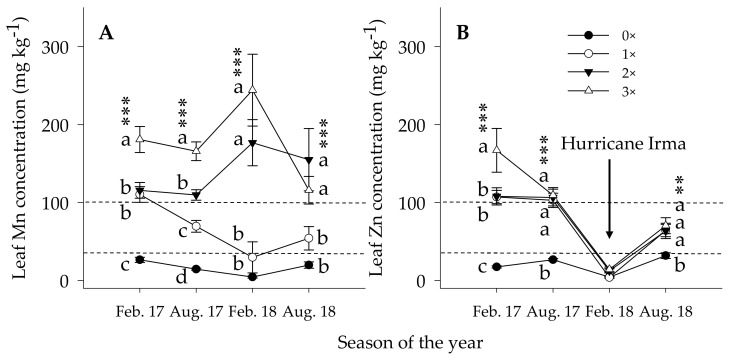
Leaf Mn (figure (**A**)) and Zn (figure (**B**)) concentration in spring (sp, Feb. /Mar) and summer (su, Aug. /Sept.) seasons of 2017–2018 growing seasons of Valencia citrus trees. Micronutrient treatments: control [0×, closed circles], foliar (1×) (1×, open circles), foliar (1×) and ground (1×) (2×, closed triangles), and foliar (1×) and ground (2×) (3×, open triangles), (1× = 9 kg ha^−1^ of metallic Mn and Zn each and 2.3 kg ha^−1^ of B). No data for trees on Volk as Hurricane Irma removed the trees in summer 2017. Asterisks: **, *** indicate significant at *p* ≤ 0.01, *p* < 0.0001, respectively. The space between the two broken lines indicates the optimum range of leaf nutrient concentrations. Means along the same vertical alignment with the same lower case letters are not significantly different (*p* ≤ 0.05) using the Tukey’s range test.

**Table 1 plants-10-00925-t001:** Soil and/or foliar applied essential nutrients on HLB-affected citrus trees at Immokalee, FL during the 2016–2018 growing seasons.

Method and Rate of Application (kg ha^−1^ year^−1^)							
	Soil	Foliar			Soil		
Treatments	N rate ^1^	Mn ^2^	Zn ^3^	B ^4^	Mn ^5^	Zn ^5^	B ^6^
0×	168	–	–	–	–	–	–
	224	–	–	–	–	–	–
	280	–	–	–	–	–	–
1×	168	9	9	2.3	–	–	–
	224	9	9	2.3	–	–	–
	280	9	9	2.3	–	–	–
2×	168	9	9	2.3	9 *	9 *	2.3 *
	224	9	9	2.3	9	9	2.3
	280	9	9	2.3	9	9	2.3
3×	168	9	9	2.3	18 **	18 **	4.6 **
	224	9	9	2.3	18	18	4.6
	280	9	9	2.3	18	18	4.6

^1^ Nitrogen as NH_4_NO_3_ for split 20 times per year from Feb. to Nov. ^2^ Foliar spray 0.011 kg L^−1^ ZnSO_4_ per hectare equivalent. ^3^ Foliar spray 0.013 kg L^−1^ MnSO_4_ per hectare equivalent. ^4^ Foliar spray 0.0012 kg L^−1^ B Na_2_B_4_O_7_ per hectare equivalent. ^5^ Soil applied sulfur encapsulated 6% ZnSO_4_ and 6% MnSO_4_ of 0.16 kg and 0.32 kg per sub subplot (each plot = 0.036 hectare) for treatments 3× * and treatment 4× **, respectively. ^6^ Ground-applied equivalent to 0.15 and 0.30 kg of Na_2_B_4_O_7_ per sub subplot (each sub subplot = 0.036 hectare) for treatments 3× * and treatment 4× **, respectively.

**Table 2 plants-10-00925-t002:** Analysis of variance (ANOVA) of soil nutrient concentration on a dry weight basis under Huanglongbing-affected Valencia citrus trees at Immokalee, FL during 2017 and 2018 growing seasons.

	Spring 2017					Summer 2017				
	NO_3_-N	NH_4_-N	Mn	Zn	B	NO_3_-N	NH_4_-N	Mn	Zn	B
Model Variables ^1^	Significance Level ^2^									
D	*	*	^NS^	^NS^	*	*	^NS^	*	*	***
N	^NS^	^NS^	^NS^	^NS^	^NS^	^NS^	^NS^	^NS^	^NS^	^NS^
M	^NS^	^NS^	*	^NS^	^NS^	^NS^	^NS^	^NS^	^NS^	^NS^
	Spring 2018					Summer 2018				
D	^NS^	*	*	^NS^	***	*	^NS^	*	^NS^	*
N	^NS^	^NS^	^NS^	^NS^	^NS^	^NS^	^NS^	^NS^	^NS^	^NS^
M	^NS^	^NS^	***	^NS^	^NS^	^NS^	*	^NS^	^NS^	^NS^
D × M	*	^NS^	^NS^	^NS^	^NS^	*	^NS^	**	**	^NS^
N × R × M	^NS^	^NS^	^NS^	^NS^	*	^NS^	^NS^	^NS^	^NS^	^NS^

^1^ Model variables: D = depths of the soil, N = nitrogen rate, and M = micronutrients. ^2^ NS, *, **, and *** represent non-significant or significant at *p* ≤ 0.05, < 0.01, and < 0.0001, respectively.

**Table 3 plants-10-00925-t003:** Effect of split soil and /or foliar-applied essential nutrients on soil nutrient concentration in 0–15 cm soil depth under Huanglongbing-affected Hamlin citrus trees at Immokalee, FL during 2017 and 2018 growing seasons.

	Soil Nutrient Concentration (mg kg^−1^)								
	Spring 2018								
Treatments ^1^	NO_3_-N	NH_4_-N	P	K	Ca	Mg	B	Cu	Fe
	**Spring 2018**								
0×	1.9 ± 1.4	3.85 ± 0.6	28 ± 4	28 ± 3	356 ± 38a	28 ± 4a	0.4 ± 0.0ab	13 ± 1	69 ± 12
1×	2.7 ± 1.5	3.72 ± 0.9	47 ± 8	27 ± 2	184 ± 22ab	21 ± 2b	0.4 ± 0.1a	12 ± 2	44 ± 7
2×	2.6 ± 1.8	5.45 ± 1.1	33 ± 5	26 ± 2	109 ± 18ab	11 ± 1b	0.3 ± 0.0b	11 ± 1	46 ± 7
3×	4.1 ± 1.7	5.40 ± 0.5	26 ± 6	18 ± 3	186 ± 73b	7 ± 2b	0.3 ± 0.0ab	11 ± 1	62 ± 8
Sign. ^2^	^NS^	^NS^	^NS^	^NS^	**	**	*	^NS^	^NS^
	**Summer 2018**								
0×	1.6 ± 0.4	3.3 ± 0.4b	21 ± 4	15 ± 1	325 ± 45a	13 ± 3a	0.2 ± 0.1	9 ± 2	59 ± 1
1×	1.7 ± 0.6	3.6 ± 0.5b	26 ± 4	18 ± 2	224 ± 37ab	13 ± 2a	0.3 ± 0.0	7 ± 1	35 ± 1
2×	1.4 ± 0.3	5.4 ± 0.8b	27 ± 3	22 ± 4	208 ± 40ab	17 ± 7a	0.2 ± 0.0	9 ± 1	55 ± 6
3×	1.2 ± 0.2	7.6 ± 1.1a	48 ± 13	12 ± 2	93 ± 27b	4 ± 3b	0.3 ± 0.1	11 ± 2	42 ± 1
Sign. ^2^	^NS^	***	^NS^	^NS^	**	***	^NS^	^NS^	^NS^

^1^ Treatments: control (0×), foliar (1×) (1×), foliar (1×) and ground (1×) (2×), and foliar (1×) and ground (2×) (3×), (1× = 9 kg ha^−1^ of metallic Mn and Zn each and 2.3 kg ha^−1^ of B). Mean soil nutrient concentration ± standard error on the same column with the same letters are not significantly different at *p* ≤ 0.05 using Tukey’s range test (*n* = 12 trees). ^2^ NS, *, **, *** Non-significant or significant at *p* ≤ 0.05, 0.01, < 0.0001, respectively.

**Table 4 plants-10-00925-t004:** Effect of rootstock and ground-applied plant nutrition on the leaf area index (LAI) and analysis of variance (ANOVA) of Huanglongbing (HLB)-affected citrus trees cv. “Valencia” budded on Volkameriana (Volk) or Swingle (Swc) rootstocks at Immokalee, FL during the 2017–2018 growing seasons.

	2017				2018				2019	
	Spring		Summer		Spring		Summer		Spring	Summer
	Leaf Area Index									
Treatments ^1^	Volk	Swc	Volk	Swc	Volk	Swc	Volk	Swc	Swc	Swc
0×	2.6	3.4	2.6	2.4	nd ^2^	3.6ab	nd	3.7a	3.4a	3.5a
1×	3.0	3.7	2.5	2.6	nd	3.7ab	nd	3.9a	3.6a	3.9a
2×	2.5	3.7	2.4	2.8	nd	3.9a	nd	3.9a	3.1a	4.0a
3×	2.7	3.5	2.2	2.3	nd	3.3b	nd	3.1b	2.3b	2.6b
Sign. ^3^	^NS^	^NS^	^NS^	^NS^	–	*	–	***	***	***
Model variables ^4^	**ANOVA**									
R	***		^NS^		^NS^		^NS^		^NS^	^NS^
M	^NS^		^NS^		*		***		***	***
N	^NS^		^NS^		^NS^		^NS^		^NS^	^NS^

^1^ Treatments: control, (0×) foliar (1×), foliar (1×) and ground (1×), and foliar (1×) and ground (2×), (1× = 9 kg ha^−1^ of metallic Mn and Zn and 2.3 kg ha^−1^ of B). Mean leaf area index on the same column with different letters are significantly different at *p* ≤ 0.05 according to Tukey’s range test (*n* = 12 trees). ^2^ No data for trees budded on Volk rootstock in 2018, as Hurricane Irma removed trees budded on Volkameriana rootstocks in September 2017. ^3^ NS, *, *** Non-significant or significant at *p* ≤ 0.05, < 0.0001, respectively. ^4^ Model variables: R = Rootstocks, N = nitrogen rate, and M = Micronutrients.

**Table 5 plants-10-00925-t005:** Effect of rootstock and ground-applied plant nutrition on the tree canopy volume and ANOVA of Huanglongbing (HLB)-affected citrus trees cv. “Valencia” budded on Volkameriana (Volk) or Swingle (Swc) rootstocks at Immokalee, FL during the 2017–2019 growing seasons.

	2017				2018				2019	
	Spring		Summer		Spring		Summer		Spring	Summer
	Tree Canopy Volume (m^3^)									
Treatments ^1^	Volk	Swc	Volk	Swc	Volk	Swc	Volk	Swc	Swc	Swc
0×	23.0b	18.4	20.2	13.6	– ^2^	14.4	–	14.7	17.6b	18.7b
1×	27.0a	18.4	23.8	13.6	–	14.3	–	17.5	19.3ab	20.2ab
2×	25.1ab	19.1	22.1	14.4	–	15.6	–	17.8	20.7a	22.6a
3×	26.3ab	18.9	22.9	14.2	–	15.0	–	16.6	19.1ab	19.5ab
Significance ^3^	*	^NS^	^NS^	^NS^	^NS^	^NS^	^NS^	^NS^	*	*
Model variables ^4^	**ANOVA**									
R	***		***		^NS^		^NS^		^NS^	^NS^
M	^NS^		^NS^		^NS^		^NS^		*	*
N	^NS^		^NS^		^NS^		^NS^		^NS^	^NS^
R×N	*		*		^NS^		^NS^		^NS^	^NS^

^1^ Model variables: control, (0×) foliar (1×), (T3) foliar (1×) and ground (1×), and (T4) foliar (1×) and ground (2×), (1× = 9 kg ha^−1^ of metallic Mn and Zn and 2.3 kg ha^−1^ of B). Mean tree canopy volume on the same column with different letters are significantly different at *p* ≤ 0.05 according to Tukey’s range test (*n* = 12 trees). ^2^ No data for trees budded on Volk rootstock in 2018, as Hurricane Irma removed trees budded on Volkameriana rootstocks in September 2017. ^3^ NS, *, *** Non-significant or significant at *p* ≤ 0.05, 0.0001, respectively. ^4^ Model variables: R = Rootstocks, N = nitrogen rate, and M = Micronutrients.

## Data Availability

not applicable.

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
