# Peer review of "The Effect of Foliar and Ground-Applied Essential Nutrients on Huanglongbing-Affected Mature Citrus Trees"

_plants, 2021, doi:10.3390/plants10050925_

Round 1
Reviewer 1 Report
I draw the authors' attention to some shortcomings of the article that must be remedied before its publication, so as not to discredit a quality journal such as this one.
The manuscript was prepared with little care and, therefore, must be improved by the authors.
More rigor is needed in the scientific names of the plants used, which have been reclassified.
There is text that is repeated in at least two cases, with different quotes in each case.
There is disagreement between the title of the article, the summary, the objectives, and the results.
Essential nutrients ≠ essential micronutrientes;
nitrogen + Zn+B ≠ N + Mn + Zn + B
There are several shortcomings of the article noted in the pdf file.

Author Response
Dear reviewer,
Please, find herewith edits to the reviews given by you. I tried to explain the highlighted part for your review. I am more than glad to accept if more comments still need to address.

Reviewer 2 Report
The hypothesis on which the study was based on lacks... In addition, which is the background? you should complete it in the Abstract (1-2 lines in the beggining).
In paragraph 2.1. it is not clear how the treatments for all nutrients were performed. Please, clarify this crucial information for the reader.
The way the results are presented are too monotonous for the readers; for example, it is completely unnecessary to indicate continuously the insignificant differences among the treatments; just focus on the significant effects and decrease the length of the Results and Discussion by at least 30%.
In addition, which is the real need to present all these results? which are the practical agronomic conclusions from all these data? for example, why do you show so many data for root nutrient concentrations?
Conclusions should be rewritted, by clearly indicating the agronomic importance of the most important data. Monotonous repetitions without meaning should be avoided.
Recommendation to the Editor: The ms needs major reconstruction in all parts before reconsider it for publication. All the ambiguities, uncertain/unclear points and monotonous information should be clarified, justified and removed, respectively. Otherwise, the paper can not ne published.
Author Response
Dear reviewer,
Please, find herewith edits to the reviews given by you. I tried to explain the highlighted part for your review. I am more than glad to accept if more comments still need to address.
With best regards,
Atta, Alisheikh Adem

Reviewer 3 Report
Although the paper is not particularly innovative, it fits well with the scope of the Journal.
I commented it, especially for the introductive part, methods and conclusions, where most of the issues reside. The Authors can find my comments in the attached pdf (tracking mode).
There are several issues regarding English grammar and syntax, so an extensive language revision is required.
A clear hypothesis was not stated in the introduction. The parts regarding diseases is not clear, as also the microorganisms involved and their names.
Generally, the parts regarding the methodology and the presentation of the results are well organized, even if sometimes details are missing, especially for the part regarding metal analysis and the specification of the nutrients' treatments.
The description of the results is sometimes hard to follow due to the various treatments adopted. It should be shortened and more links to tables and figures provided.
The conclusions are interesting, especially from an applicative point of view. Considering that the manuscript is quite descriptive, while mechanistic or physiological aspects are not fully covered, the conclusions should be rewritten, highlighted and valorised.
I suggest the eventual publication of the paper after major revision focused on both the form and the content of the article. More details are given in the attached pdf.

Author Response
Dear reviewer,
I addressed the edits suggested attached to this e-mail. I am also committed to add or edit more comments if you have any suggestions. Thank you.
With kind regards,
Atta, Alisheikh Adem

Round 2
Reviewer 3 Report
The Authors answered to all the points raised and edited the manuscript accordingly. In particular, the hypotheses and objectives of the study were clarified. Microbial nomenclature has been corrected. Tha authors also explained that no mineralization process was undertaken in the course of the soil nutrient analysis, clarifying another point.
On this basis, although the manuscript is not particularly sound nor innovative, it reached an average evaluation that allows the publication in Plants.
Author Response
Dear reviewer,
Thank you for the comments and appreciate it. Please find herewith the comments as requested for improvement.
With best regards,
On behalf of the authors
